# Structure of the human plasma membrane $Ca^{2+}$-ATPase 1 in complex with its obligatory subunit neuroplastin

Deshun Gong [1], Ximin Chi[1], Kang Ren[1], Gaoxingyu Huang[1], Gewei Zhou[1], Nieng Yan[1,2], Jianlin Lei [3] & Qiang Zhou [4]

Plasma membrane $Ca^{2+}$-ATPases (PMCAs) are key regulators of global $Ca^{2+}$ homeostasis and local intracellular $Ca^{2+}$ dynamics. Recently, Neuroplastin (NPTN) and basigin were identified as previously unrecognized obligatory subunits of PMCAs that dramatically increase the efficiency of PMCA-mediated $Ca^{2+}$ clearance. Here, we report the cryo-EM structure of human PMCA1 (hPMCA1) in complex with NPTN at a resolution of 4.1 Å for the overall structure and 3.9 Å for the transmembrane domain. The single transmembrane helix of NPTN interacts with the $TM_{8-9}$-linker and TM10 of hPMCA1. The subunits are required for the hPMCA1 functional activity. The NPTN-bound hPMCA1 closely resembles the E1-$Mg^{2+}$ structure of endo(sarco)plasmic reticulum $Ca^{2+}$ ATPase and the $Ca^{2+}$ site is exposed through a large open cytoplasmic pathway. This structure provides insight into how the subunits bind to the PMCAs and serves as an important basis for understanding the functional mechanisms of this essential calcium pump family.

[1] Beijing Advanced Innovation Center for Structural Biology, Tsinghua-Peking Joint Center for Life Sciences, School of Life Sciences, Tsinghua University, Beijing 100084, China. . [2] Present address: Department of Molecular Biology, Princeton University, Princeton, NJ 08544, USA. [3] Technology Center for Protein Sciences, Ministry of Education Key Laboratory of Protein Sciences, School of Life Sciences, Tsinghua University, Beijing 100084, China. [4] Beijing Advanced Innovation Center for Structural Biology, Tsinghua-Peking Joint Center for Life Sciences, School of Medicine, Tsinghua University, Beijing 100084, China. These authors contributed equally: Deshun Gong, Ximin Chi, Kang Ren. Correspondence and requests for materials should be addressed to D.G. (email: dsgong@tsinghua.edu.cn) or to Q.Z. (email: zhouqiang00@tsinghua.org.cn)

Tight regulation of $Ca^{2+}$ signaling is crucial for cell function and survival. The plasma membrane $Ca^{2+}$ ATPase (PMCA) plays an essential role to regulate cellular $Ca^{2+}$ homeostasis in all eukaryotic cells. PMCA extrudes excess $Ca^{2+}$ from the cytoplasm, a process that maintains a steep gradient between intracellular (~100 nM) and extracellular $Ca^{2+}$ (~2 mM)[1,2]. In nonexcitable cells where the resting-state $Ca^{2+}$ concentration remains low, PMCA is generally the principal $Ca^{2+}$ clearance system[3,4]; In excitable cells such as myocytes and neurons with higher demand for $Ca^{2+}$ clearance, PMCA cooperates with the sodium/calcium exchanger (NCX) and endo(sarco)plasmic reticulum $Ca^{2+}$ ATPase (SERCA) in the global maintenance of cellular $Ca^{2+}$ homeostasis[5,6]. In addition, the importance of PMCA in the regulation of local intracellular $Ca^{2+}$ dynamics has steadily increased. It generates a microdomain in its vicinity with low $Ca^{2+}$ concentration, thereby negatively regulating $Ca^{2+}$-dependent interaction partners by attracting them to its locale in caveolae[7]. Genetic deletion or loss-of-function mutations of individual PMCAs are associated with a variety of human diseases, including cardiovascular disease, cerebellar ataxia, deafness, paraplegia, and infertility[7–10].

PMCA belongs to the family of P-type ATPases. Three $Ca^{2+}$-ATPases were identified in animal cells, the class PIIA SERCAs and golgi secretory pathway $Ca^{2+}$-ATPases (SPCAs), and the class PIIB PMCAs[6]. Although they share essential properties of membrane topology and working mechanism, PMCAs have some unique properties distinct from the other two $Ca^{2+}$-ATPases, particularly in the regulation regions. PMCAs have a unique autoinhibitory domain at N terminus (in plants) or C terminus (in mammals)[11–14]. The action of this domain can be suppressed by directly binding to calmodulin (CaM) or acidic phospholipids[15,16]. In addition, an additional phospholipid-binding domain that contains about 40 predominantly basic amino acids exists in the first cytosolic loop of mammalian PMCAs, providing the second binding site for acidic phospholipids[17]. Recently, two immunoglobulin (Ig) superfamily proteins, neuroplastin (NPTN) and basigin (BASI), were identified as previously unrecognized obligatory subunits of PMCAs and essential for the efficient control of the PMCA-mediated $Ca^{2+}$ clearance[10,18,19]. However, detailed structural information is unavailable for NPTN- or BASI-bound PMCAs and the molecular mechanism underlying the active effect of the subunits on PMCAs.

Due to their physiological and pathological significance and identification as novel complexes with NPTN and BASI, PMCAs represent important targets for structural characterization. In this manuscript, we report the cryo-electron microscopy (cryo-EM) structure of the human PMCA1 in complex with NPTN at an average resolution of 4.1 Å, with the local resolution at the transmembrane region of the complex reaching 3.9 Å. This structure offers important insight into how the NPTN binds to the PMCA1 and serves as a molecular framework for the mechanistic understanding of the efficient control of PMCA-mediated $Ca^{2+}$ clearance by NPTN.

## Results

**Structure determination of the human PMCA1–NPTN complex.** To obtain structural information on PMCAs, full-length human PMCA1d (hereinafter referred to as hPMCA1) was successfully overexpressed in mammalian HEK293F cells by transient transfection. Surprisingly, certain smear bands with molecular weights of approximately 55 kDa co-migrated with hPMCA1 on size-exclusion chromatography (Supplementary Fig. 1a). An analysis of 2D class averages showed that an additional density that does not belong to hPMCA1 is present at the

extracellular side (Supplementary Fig. 1b), suggesting that hPMCA1 is complexed with an endogenous glycosylated protein. Mass spectrometric (MS) analyses of the additional bands identified both NPTN and BASI (Supplementary Table 1). As a method for isolating hPMCA1-NPTN from hPMCA1-BASI is not available, the density of the subunit in our structure may belong to both NPTN and BASI. We proceeded with cryo-electron microscopy (cryo-EM) analysis under the assumption that the high sequence conservation of the two proteins could be used to resolve the structure of most of the functional elements. Fortunately, the functional residues discussed in this study are invariant between the two proteins. For descriptive purposes, we will refer to this structural complex as NPTN in the remainder of this work.

The reconstruction shows an hPMCA1 molecule associated with an NPTN molecule in our structure. The overall resolution of the EM map is calculated to 4.1 Å using 105,118 selected particles according to the gold-standard Fourier shell correlation (FSC) 0.143 criterion (Fig. 1a, b, Supplementary Table 2, and Supplementary Fig. 2). The EM map is well resolved for most of the P domain and the transmembrane domain, where the local resolution reaches 3.9 Å (Fig. 1a). The main chain of these regions was built by homology modeling based on the crystal structure of SERCA (PDB: 3W5B) and the side chains were assigned mainly by bulky residues such as Phe, Tyr, Trp, and Arg (Supplementary Fig. 4a). The densities for the A domain and the N domain were of lower resolutions. Predicted structures for these two domains generated in Phyre2[20] can be docked into the map with minor adjustment (Fig. 1a and Supplementary Fig. 4b). In a low-pass-filtered EM map at 6.0 Å resolution, the orientation of the Ig-domain 2 (Ig-2) can be reliably determined, thereby allowing for docking of the crystal structure of the Ig-2 into the map (Supplementary Fig. 4c). However, the density of the Ig-1 is largely missing. In this paper, the structural elucidation is mainly focused on the transmembrane domain with high resolution.

**The NPTN-TM interacts with the TM$_{8-9}$-linker and TM10.** The domain organization of hPMCA1 closely resembles that of other P-type ATPases and consists of three large cytoplasmic domains (A, actuator; N, nucleotide binding; P, phosphorylation) and ten transmembrane helices (TM1-10) (Fig. 1c). The C-terminal autoinhibitory domain and the phospholipid-binding domain[17] in the first cytosolic loop of the PMCAs are not resolved, suggesting structural flexibility in these regions. The NPTN subunit resembles a gun wherein the TM and Ig-domains form the handle and barrel, respectively (Supplementary Fig. 4c). The NPTN-TM traverses the membrane with a tilt angle of approximately 30° (Fig. 1c). It is positioned adjacent to the TM10 and far from the TM1-9 transmembrane helices of hPMCA1. The NPTN-TM and TM10 of hPMCA1 show intimate interactions through a number of hydrophobic residues near the extracellular surface of the membrane and are far away from each other at the intracellular end. The TM$_{8-9}$-linker serves as an anchor that stabilizes the interaction (Fig. 2a and Supplementary Fig. 5a). These contact residues are invariant between NPTN and BASI, suggesting that these two proteins share the same binding surface with PMCAs (Fig. 2b). The TM$_{7-8}$-linker of hPMCA1 may be responsible for the binding to Ig-2 of NPTN (Supplementary Fig. 5b).

To our knowledge, the binding surface shown here is unique among the known interactions of P-type ATPases with their subunits and modulators. Previous structural information on multi-subunit P-type ATPases was obtained in studies of the $Na^+$, $K^+$-ATPase β and γ subunits[21] and the $H^+$, $K^+$-ATPase β subunit[22,23]. The β subunits of these ATPases make direct contact with TM7 and TM10, and the γ-subunit is next to TM9.

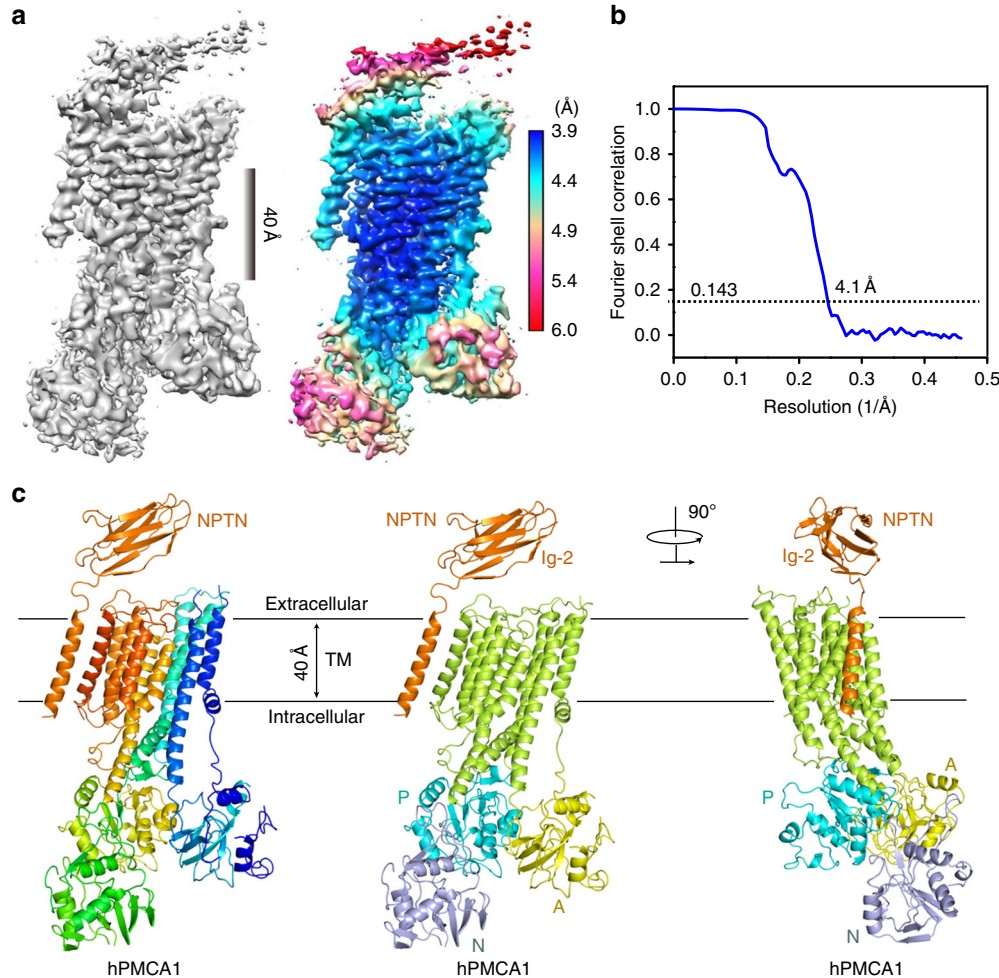

**Fig. 1** Cryo-EM structure of hPMCA1 in complex with the NPTN subunit. **a** Overall EM map of the hPMCA1–NPTN complex. The image of the 4.1-Å map was generated in Chimera[60]. The density of Ig-2 is not visible at this threshold. Right panel: Local resolution map estimated with RELION 2.0[54]. **b** Gold-standard Fourier shell correlation curve for the cryo-EM map. **c** Overall structure of the hPMCA1–NPTN complex. The structure on the left is colored in rainbow with the amino and carboxyl termini colored blue and red, respectively. The structures of hPMCA1 on the middle and right are domain colored, and the NPTN subunit is shown in orange. The same color scheme is used throughout the manuscript. All structural figures were prepared using PyMol (http://www.pymol.org)

Similarly, the accessory regulatory protein FXYD10 also interacts almost exclusively with TM9[24] (Fig. 2c). Additional structural information on the interaction of P-type ATPases with their modulators was obtained from studies of the SERCA-SLN (sarcolipin) complex[25,26]. SLN was shown to associate with SERCA through a groove surrounded by TM2, TM6, and TM9 (Fig. 2d). Taken together, the hPMCA1–NPTN complex structure represents a novel binding pattern between P-type ATPases and their subunits or modulators.

It has been reported that the NPTN–PMCA interaction was sensitive to solubilization conditions[10]. To investigate the role of the subunits in the regulation of the hPMCA1 functional activity, detergent screening was performed during the purification to obtain the hPMCA1 alone proteins. The complex was dissociated by washing with dodecyltrimethylammonium chloride (DTAC)-containing buffer (Fig. 2e). Most of the hPMCA1 alone proteins were still well folded (Supplementary Fig. 6). Accordingly, the ATPase activities of purified hPMCA1-NPTN and hPMCA1 alone proteins were examined. The $K_m$ and $V_{max}$ for the ATPase activity of hPMCA1-NPTN proteins were measured to be ~519.5 μM and ~ 325.5 nmol mg$^{-1}$ min$^{-1}$, respectively. The hPMCA1 alone proteins were devoid of ATPase activity (Fig. 2f). These results indicate that the hPMCA1-NPTN proteins are functional and the subunits are required for the hPMCA1 functional activity.

**The hPMCA1 closely resembles the E1-Mg$^{2+}$ structure**. The E2-E1 equilibrium of PMCAs is shifted more towards the E2 conformation in the presence of EDTA[2]. To trap the protein in the autoinhibited state, 5 mM EDTA was added to the buffer in the last step of purification. However, the structure of the NPTN-bound calcium pump differs from the E2 conformation of SERCA (root mean squared deviation (r.m.s.d.) ~7.5 Å) and more closely resembles the E1-Mg$^{2+}$ conformation (r.m.s.d. ~3.0 Å) (Fig. 3a). The TM1 is sharply bent in hPMCA1, very similar to that in E1-Mg$^{2+}$ structure; the TM2, TM3, TM5, TM6, TM8, and TM9 in hPMCA1 are well aligned with those in E1-Mg$^{2+}$ structure. Conspicuous differences are observed in TM1, TM4, TM7, and TM10. To facilitate the binding of NPTN-TM, TM7, and TM10 show dramatic movement towards NPTN-TM. The region near the extracellular ends of TM1 and TM4 also shows obvious movement towards TM2 that may be induced by the rearrangement of the Ca$^{2+}$-binding pocket (Fig. 3b). When the two structures are superimposed in the transmembrane domain, each of the three cytoplasmic domains shows obvious global

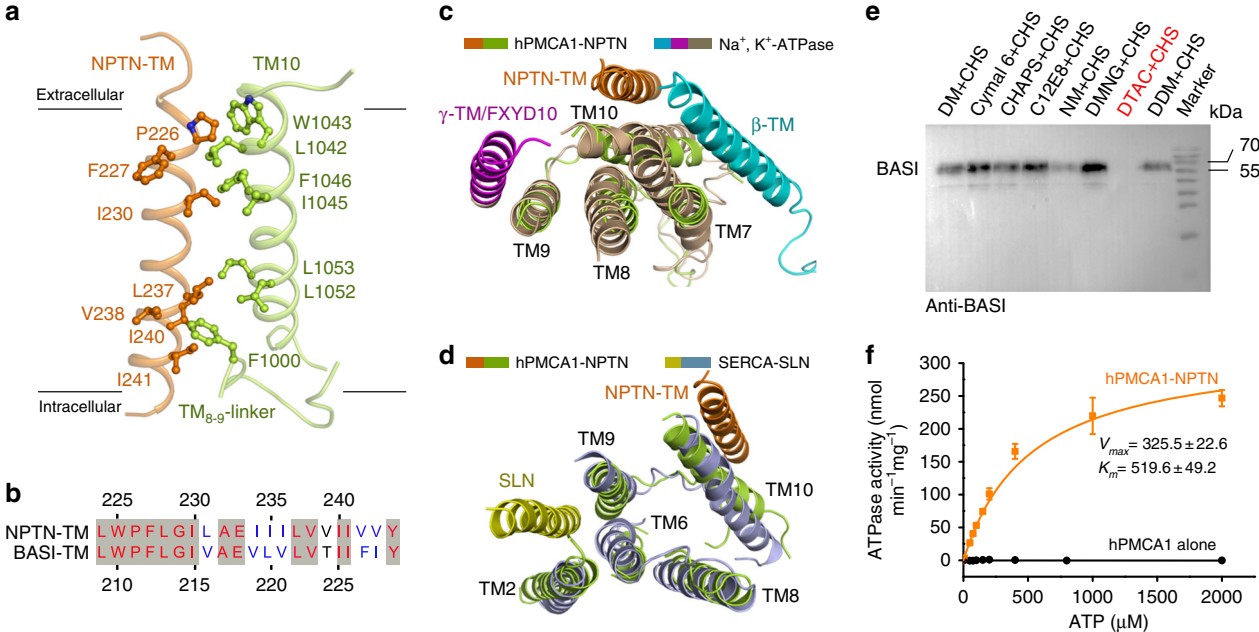

**Fig. 2** Interactions between the transmembrane regions of hPMCA1 and NPTN subunit. **a** NPTN-TM interacts with TM10 and the TM$_{8-9}$-linker of hPMCA1. The hydrophobic residues on the interface are shown. **b** Sequence alignment of NPTN-TM and BASI-TM. **c** Structural comparison of the NPTN-TM binding site on hPMCA1 with that of β-TM and γ-TM/FXYD10 on Na$^+$, K$^+$-ATPase (PDB: 4HQJ). The α-subunit of Na$^+$, K$^+$- ATPase is shown in light brown, the β-TM is shown in cyan, and the γ-TM/FXYD10 is shown in magenta. The structure is viewed from the extracellular side. **d** Structural comparison of the NPTN-TM binding site on hPMCA1 with that of the SLN on SERCA (PDB: 4H1W). SERCA is shown in light blue, and the SLN is shown in yellow. The structure is viewed from the extracellular side. **e** Detergent screening for obtaining the hPMCA1 alone proteins. The complexes of hPMCA1-subunits fell apart by washing with DTAC-containing buffer. DM n-decyl-alpha-D-maltopyranoside, DMNG decyl maltose neopentyl glycol, NM n-nonyl-beta-D-maltopyranoside, DDM n-dodecyl-beta-D-maltopyranoside, C12E8 octaethylene glycol monododecyl ether, DTAC dodecyltrimethylammonium chloride, Cymal 6 6-cyclohexyl-1-hecyl-beta-D-Maltoside. **f** Measurement of ATPase activities of the hPMCA1-NPTN and hPMCA1 alone proteins. Each data point is the average of three independent experiments and error bars represent SD

movement compared with its position in the E1-Mg$^{2+}$ conformation (Fig. 3c). These results indicate that the structure of NPTN-bound hPMCA1 closely resembles the E1-Mg$^{2+}$ structure of SERCA.

**Ca$^{2+}$-binding site and access channel.** Compared with the E1-2Ca$^{2+}$ structure, the Ca$^{2+}$-binding site of hPMCA1 is formed by E433 in TM4 and by D895, N891 in TM6, and this site is highly conserved with the Ca$^{2+}$-binding site II. The Ca$^{2+}$-binding site I is not preserved in PMCAs due to substitution of the essential acidic residue E771 in TM5 and E908 in TM8 of SERCA by A866 and Q983 in hPMCA1 (Fig. 4a, b), respectively. Similar to the E1-Mg$^{2+}$ conformation of SERCA, a large open mouth was formed by the TM1 kink, TM2, TM3, and TM4 near the cytoplasmic surface of the membrane extends towards the transmembrane Ca$^{2+}$-binding site (Fig. 4c). The electrostatic potential surface shows that the Ca$^{2+}$ permeation pathway is funnel shaped and consists of a large cytosolic vestibule leading to a narrow transmembrane tunnel. Numerous negatively charged residues (E104, D108, D174, and E178) are present in the funnel, thereby contributing to cation selectivity (Fig. 4d). Accordingly, the E1-NPTN structure shown here represents an E1-Mg$^{2+}$-like intermediate conformation between E2 and E1-Ca$^{2+}$; in this conformation, the Ca$^{2+}$-binding site is exposed to the cytoplasm and ready to accept new cytosolic Ca$^{2+}$.

**TM1 sliding door of hPMCA1.** A TM1 sliding door in SERCA and Na$^+$, K$^+$-ATPase control the exposure of the cation-binding site to the cytoplasm[25,27]. For instance, the TM1 of SERCA is sharply bent as a TM1 kink, with the hydrophobic residue L61 of TM1 and the small residue G257 of TM3 serving as pivot points.

The conserved L65 of TM1 functions as a gate-lock residue that restricts the mobility of the side chain of E309 in TM4, a key residue for Ca$^{2+}$ binding and release. Compared with the E2 state of SERCA, T110 of TM1 and A370 of TM3 serve as pivot points for the kink in hPMCA1, whereas L114 restricts the mobility of E433. Notably, compared with the SERCA(E2) conformation, the TM1' of hPMCA1-NPTN occupies a significantly higher position with respect to the membrane. The distance between the Cα atoms of T110 in hPMCA1-NPTN and L61 in SERCA(E2) can be as high as 11 Å, indicating that significant movement of the TM1 sliding door in E1-NPTN occurs to expose the Ca$^{2+}$-binding site (Fig. 5a). The position of the TM1 kink is similar to that observed in the E1-Mg$^{2+}$ state of SERCA, in which T110 faces L427 of TM4 and L114 associates with V424 of TM4. In the E2 state of SERCA, in which the Ca$^{2+}$ entry pathway is blocked, the distance between the Cα atoms of G257 and L61 is 6 Å. Correspondingly, the distance between the Cα atoms of A370 and T110 in hPMCA1-NPTN increases to 16 Å (Fig. 5a, b). Accordingly, the Ca$^{2+}$ entry pathway becomes accessible. A cartoon is presented in Fig. 5c to illustrate the exposure of the Ca$^{2+}$-binding site through sliding of TM1 during the transition from the E2 state to the E1 state.

**Discussion**

P-type ATPases are fundamental in establishing and maintaining steep gradients of key cations across membranes. The P-type ATPase superfamily encompasses 11 distinct classes, covering a wide range of cationic and lipid substrates[28,29]. Members of the class PIIC (Na$^+$, K$^+$-ATPase and H$^+$, K$^+$-ATPase) and most of the PIV subfamily ATPases form a heterocomplex with at least one additional subunit, which is essential for function[30]. Only

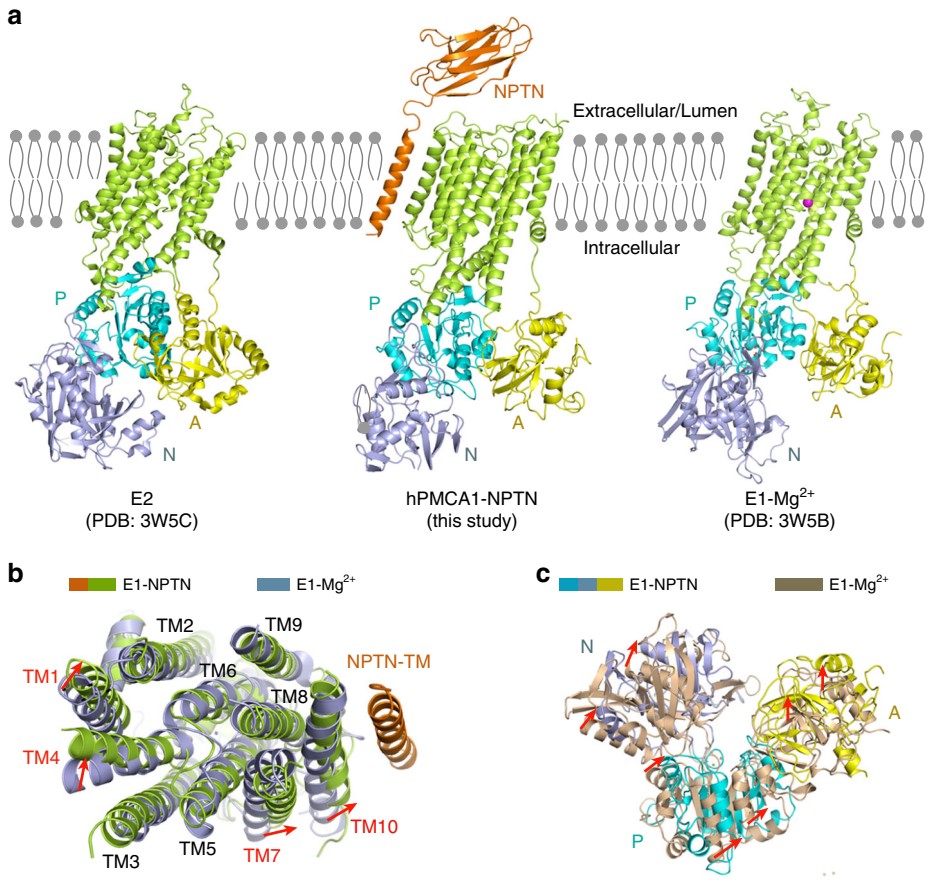

**Fig. 3** NPTN-bound calcium pump is in an E1-Mg$^{2+}$-like state. **a** Structural comparison of hPMCA1-NPTN with the E2 (PDB: 3W5C) and E1-Mg$^{2+}$ (PDB: 3W5B) conformations of SERCA. **b** Conformational changes in the transmembrane regions of E1-NPTN and E1-Mg$^{2+}$. The red arrows indicate the shifts in the corresponding elements from E1-Mg$^{2+}$ to E1-NPTN. **c** Conformational changes in the cytoplasmic domains of E1-NPTN and E1-Mg$^{2+}$; the two structures are superimposed relative to the transmembrane domain

recently, the class PIIB PMCAs were identified as heteromeric complexes that are assembled from two ATPases and two essential auxiliary subunits, either NPTN or BASI[10], instead of being monomers or homodimers as previously envisaged[9,15,31]. Atomic structures of P-type ATPases have been determined for the class PIB copper-transporting ATPase[32] and zinc-transporting ATPase[33], the class PIIA SERCA[34], the class PIIC Na$^+$, K$^+$-ATPase[21] and H$^+$, K$^+$-ATPase[23], and the class PIIIA H$^+$-ATPase[35]. In this manuscript, the structure of a PIIB Ca$^{2+}$-ATPase in complex with its obligatory subunit marks an important step towards understanding the functional mechanisms of this essential calcium pump family.

Our structure offers the first picture on the molecular appearance of PMCAs. The reconstruction shows an hPMCA1 molecule associated with an NPTN molecule in our structure (Fig. 1). The native PMCAs are assembled as heterotetramers of two ATPase subunits and two NPTN or BASI molecules[10], suggesting that the quaternary complex may be dissociated in the detergent environment. Interestingly, the hPMCA1 alone proteins were devoid of ATPase activity (Fig. 2f). It has been reported that the PMCA-mediated Ca$^{2+}$ transport was largely abolished in the NPTN/BASI double knockout cells, an effect comparable with that of washout of ATP[10]. Transient expression of PMCA2 led to only partial restoration of Ca$^{2+}$ transport, indicating that Ca$^{2+}$ may be transported by PMCA2 alone in vivo[10]. A possible explanation for this difference is that the lipids in the plasma membrane play important roles in regulating the activity of PMCAs[36].

The residues 206–271 (A domain) and 537–544 (N domain) of hPMCA4 serve as two receptor sites for interacting with the CaM-binding sites (CaM-BS) of autoinhibitory domain[11,12]. The access of proteases to their cleavage sites near the CaM-BS was used as a measure of regulatory interaction in PMCAs. The cleavage sites are completely protected in the presence of EDTA, indicating that the CaM-BS tightly interacts with the receptor sites and the E2-E1 equilibrium is shifted more toward the E2 conformation[2]. The structure of NPTN-bound hPMCA1 closely resembles the E1-Mg$^{2+}$ intermediate even in the presence of EDTA (Fig. 3), with exposure of the Ca$^{2+}$ site through an open cytoplasmic pathway (Fig. 4c, d), indicating that the NPTN may improve the efficiency of PMCA-mediated Ca$^{2+}$ transport by facilitating the transition of hPMCA1 from E2 to E1 conformation.

However, the molecular mechanism for the transition from the E1-NPTN state to the autoinhibited state remains unknown. As the NPTN is required for the hPMCA1 functional activity (Fig. 2f), we speculate that the transition may be accompanied by the dissociation of subunits from the PMCAs in the plasma membrane. Nevertheless, there is a possibility that, lipids of plasma membrane could influence this process in native environment. The PMCA activity is influenced by the phospholipid composition in the surrounding plasma membrane[36,37]. Acidic phospholipids and polyunsaturated fatty acids activate the pump by binding to two sites in the pump: one is the CaM-BS[17], the other is the phospholipid-binding domain in the cytosolic loop that connects TM2 and TM3[38]. Structure analysis indicates that

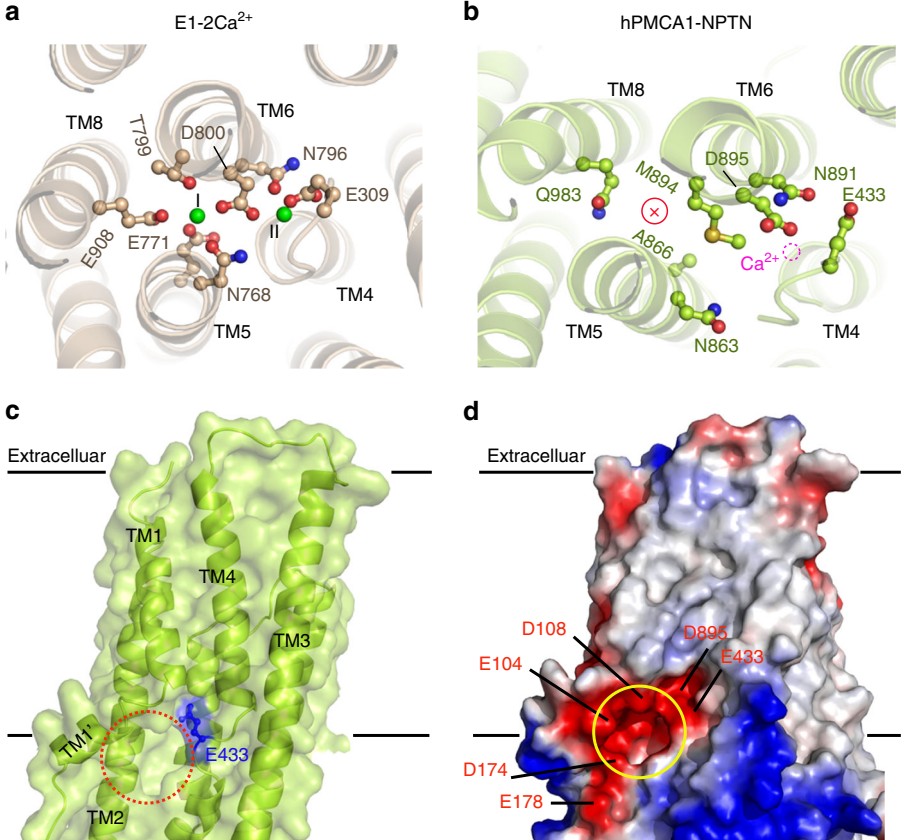

**Fig. 4** Ca$^{2+}$-binding site and Ca$^{2+}$ Access channel. **a** Two Ca$^{2+}$-binding sites (green) in E1-2Ca$^{2+}$ of SERCA (PDB: 1SU4). The structure is viewed from the cytoplasmic side. **b** Single Ca$^{2+}$-binding site in hPMCA1. The magenta dashed circle represents the Ca$^{2+}$-binding site; and the capital X within the red circle represents the missing first Ca$^{2+}$-binding site. The structure is viewed from the cytoplasmic side. **c** Surface representation of the Ca$^{2+}$-binding site and the access channel. **d** Electrostatic properties of the interior surfaces of the Ca$^{2+}$ access pathways of E1-NPTN. The negatively charged residues are highlighted

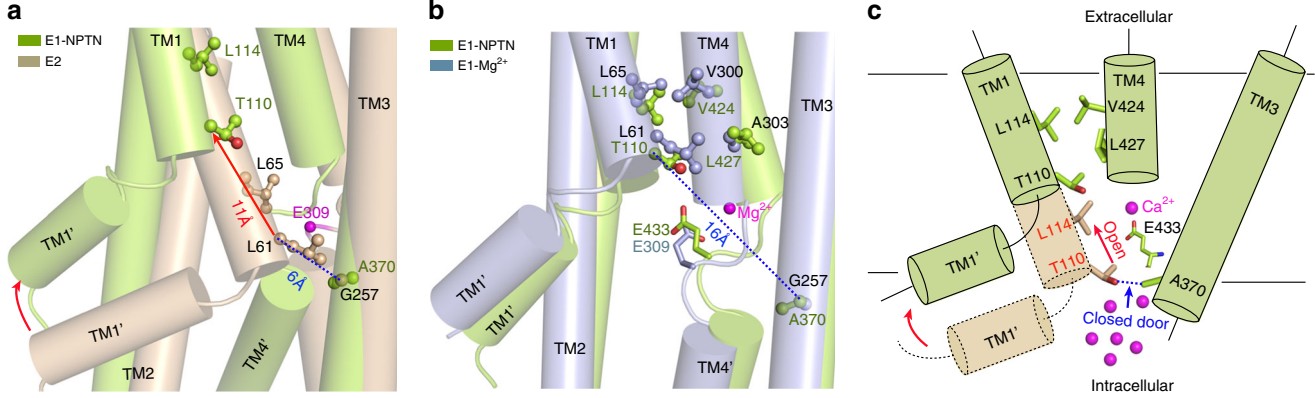

**Fig. 5** TM1 sliding door controls the exposure of the Ca$^{2+}$-binding site. **a** TM1 sliding door of E1-NPTN is open compared with its position in the E2 state. The two structures are superimposed relative to TM3. The red arrows indicate the shifts of the corresponding elements from the E2 state to the E1-NPTN state. E2 is shown in light brown. **b** Structural similarity of the TM1 sliding door in the E1-NPTN and E1-Mg$^{2+}$ states. E1-Mg$^{2+}$ is shown in light blue. **c** Schematic illustration of the structural shifts required to expose the Ca$^{2+}$-binding site in hPMCA1

the phospholipid-binding domain is located in the vicinity of the large cytosolic vestibule of Ca$^{2+}$ permeation pathway (Supplementary Fig. 7), suggesting that the phospholipid-binding domain may directly affect the Ca$^{2+}$ access channel by interacting with acidic phospholipids. The concentration of the doubly phosphorylated derivative of phosphatidyl inositol (PIP2), the most effective acidic phospholipid in stimulating PMCA activity, is modulated during Ca$^{2+}$-related signaling processes. Accordingly, a possible PIP2-mediated reversible PMCA inactivation

mechanism could be envisaged[6,39]. Structures of PMCAs in more conformations during the transport cycle are necessary to fully understand the regulatory mechanisms of the subunits and the autoinhibitory domain on PMCAs.

The structure of the hPMCA1–NPTN complex will facilitate future investigation on the pathogenic mechanism of mutations on PMCAs. The genome-wide association studies in recent years have suggested potential significance of PMCAs in human health and diseases[7]. A number of point mutations on PMCAs have

been associated with phenotypes in human and mouse[40–47]. Among the identified mutations, 5 out of 7 mutations on PMCA2, 1 out of 3 on PMCA3, and one on PMCA4 can be reliably mapped to the structure (Supplementary Figs. 3 and 8). In sum, our structural analysis provides an important framework for the elucidation of the function and disease mechanism of this essential calcium pump family.

## Methods

**Expression and purification of human PMCA1.** The complementary DNA of full-length *hPMCA1d* was subcloned into the pCAG vector (Supplementary Table 3). A C-terminal FLAG tag and a C-terminal His$_8$ tag were fused for two-step purification. HEK293F cells (Invitrogen) were cultured in SMM 293T-I medium (Sino Biological Inc.) at 37 °C under 5% $CO_2$ in a Multitron-Pro shaker (Infors, 130 rpm). When the cell density reached $2.0 \times 10^6$ cells per ml, the pCAG-PMCA1 plasmids were transiently transfected into the cells. For one-litre cell cultures, approximately 1.5 mg of plasmid was pre-mixed with 4.0 mg 25-kDa linear poly-ethylenimines (PEIs) (Polysciences) in 50 ml fresh medium for 20–30 min before transfection. The 50 ml mixture was then added to the cell culture, and the culture was incubated for 30 min for transfection. The transfected cells were cultured for 48 h before harvesting.

For purification of hPMCA1, 12 l of cells were collected and resuspended in lysis buffer containing 25 mM Tris pH 8.0, 150 mM NaCl, 1.3 µg/ml aprotinin, 1 µg/ml pepstatin, 5 µg/ml leupeptin, and 0.2 mM PMSF (lysis buffer A). The membrane fraction was solubilized at 4 °C for 2 h in 1% (w/v) *N*-dodecyl β-ᴅ-maltoside (DDM) and 0.2% (w/v) cholesterol hemisuccinate (CHS). After centrifugation at 25,000 x *g* for 40 min at 4 °C, the supernatant was passed over an anti-FLAG M2 affinity gel (Sigma) column twice. The resin was washed three times with 10 ml wash buffer A (lysis buffer A plus 0.02% DDM and 0.004% CHS). The protein was eluted with elution buffer A (wash buffer A plus 200 µg/ml FLAG peptide (Sigma)). The eluent was incubated with nickel affinity resin (Ni-NTA, Qiagen) at 4 °C for 40 min, the resin was washed with wash buffer B (lysis buffer A plus 0.1% (w/v) digitonin (Sigma) and 10 mM imidazole), and the protein was eluted with elution buffer B (lysis buffer A plus 0.1% digitonin and 300 mM imidazole). The eluent was concentrated with a 100-kDa cutoff Centricon (Millipore) and subjected to size-exclusion chromatography (SEC, Superose 6, 10/300, GE Healthcare) in a buffer containing 25 mM Tris pH 8.0, 150 mM NaCl, 1.3 µg/ml aprotinin, 1 µg/ml pepstatin, 5 µg/ml leupeptin, 0.2 mM PMSF, 0.1% digitonin, 2 mM DTT, and 5 mM EDTA. For the cryo-EM analysis, the peak fractions were concentrated to ~8 mg/ml by a 100-kDa cutoff Centricon.

To obtain the hPMCA1 alone proteins, detergent screening was performed during purification. The hPMCA1-NPTN proteins used for ATPase activity assay were purified as mentioned above. The hPMCA1 alone proteins were purified similarly, except that DDM was replaced by different detergents in washing and elution steps of the first-step purification and Superose 6 column was replaced by Superdex 200 column in the last step purification. The subunit BASI was detected by the anti-BASI antibodies (R&D Systems).

**Sample preparation and cryo-EM data acquisition.** Vitrobot Mark IV (FEI) was used in the preparation of the cryo-EM grids. Aliquots (3 µl each) of hPMCA1-NPTN protein were placed on glow-discharged Quantifoil (1.2/1.3) 300 mesh Au grids (Zhongjingkeyi Technology Co. Ltd.). The grids were blotted for 4 s and plunged into liquid ethane cooled with liquid nitrogen. The grids were then transferred to a Titan Krios (FEI) electron microscope equipped with a Gatan GIF Quantum energy filter and operated at 300 kV with a nominal magnification of 105,000×. Zero-loss movie stacks were automatically collected using AutoEMationII[48,49] with a slit width of 20 eV on the energy filter and a defocus range from −1.5 µm to −2.5 µm. Each stack was exposed in super-resolution mode for 5.6 s with an exposure time of 0.175 s per frame, resulting in 32 frames per stack. The total dose was approximately 50 e⁻/Å² for each stack. The stacks were first motion-corrected with MotionCorr[50] and binned by twofold, resulting in a pixel size of 1.091 Å/pixel. The output stacks from MotionCorr were further motion-corrected with MotionCor2[51], and dose weighting was performed[52]. The defocus values were estimated using Gctf[53].

**Image processing.** A diagram of the procedures used in data processing is presented in Supplementary Fig. 2. Approximately 3000 particles were manually picked and used to generate 2D classes for templates for auto-picking. A total of 1,730,910 particles were auto-picked from 4100 micrographs with RELION 2.0[54]. After 2D classification, ten good 2D classes were used to generate an initial model using e2initialmodel.py[55], and a total of 1,001,249 good particles were then selected and subjected to 3D auto-refinement. The particles were further subjected to several cycles of 3D classification with six classes and a local angular search step of 3.75° with the output from different global angular search iterations of the 3D auto-refinement as input. The class with fully intact particles was considered as a good class, which contains useful high-resolution information and usually has the smallest value of the accuracy of rotation and translation. A total of non-duplicated 655,998 particles were selected from the good classes of local angular search 3D

classification. These particles were subjected to local angular search 3D auto-refinement with a soft mask applied, resulting in a 4.5-Å resolution map. The particles were classified into four classes using multi-reference, and the best classes were selected and combined. The final particle number for the 3D auto-refinement is 105,118, thereby resulting in a 4.1-Å resolution map after post-processing. The resolution was estimated with the gold-standard Fourier shell correlation 0.143 criterion[56] with the high-resolution noise substitution method[57].

**Model building and refinement.** The 4.1-Å reconstruction map was used for model building. The structure of E1-Mg²⁺ (PDB: 3W5B), which was first fitted into the EM map by Chimera, served as a reference for model building. Model building was performed in COOT[58]. Bulky residues, such as Phe, Tyr, Trp, and Arg, in many of the TMs and in the P domain of hPMCA1 were clearly visible in our cryo-EM structure and used as landmarks for model building. The secondary structure predicted by Phyre2[20] based on the sequence of the A domain (residues 193–290) was well fitted into the map, and the bulky residues F194, R198, R219, and Y220, which were clearly resolved, and several motifs that are highly conserved between SERCA and PMCAs facilitated the sequence assignment (Supplementary Fig. 3). The structure formed by residues 73–89 of the N terminal region was built based on the structure of SERCA (PDB: 3W5B). Residues 2–72 of the N terminal region were built as poly-Ala due to the lack of homolog structure. The N domain predicted by Phyre2 was fitted into the map and manually adjusted in COOT; however, tracing the main chains of the β-strands was challenging due to the lower resolution. For the NPTN, the bulky residues W225 and F227 in the transmembrane domain were clearly resolved, thereby facilitating the sequence assignment. The Ig-2 of the crystal structure of rabbit NPTN (PDB: 2WV3) was fitted into the low-pass-filtered 6.0-Å resolution map, and the density of glycosylation site N168 was used for model confirmation. Modeling of the Ig-1 failed due to difficulty in determining its orientation at the low-pass-filtered 6.0-Å resolution. Structure refinement was performed by PHENIX[59] in real space with a secondary structure and geometry restraints. The statistics of the 3D reconstruction and model refinement are summarized in Supplementary Table 2.

**ATPase activity assay.** The hPMCA1-NPTN and hPMCA1 alone proteins used for ATPase activity assay were purified as described above. The ATPase activity was measured using QuantiChrom ATPase/GTPase assay kit (BioAssay Systems). The protein concentrations for the assays ranged from 0.05 to 0.2 mg/ml. All reactions were performed using the reaction buffer from the assay kit with final concentration of 1.83 mM CaCl$_2$, 5 mM MgCl$_2$, 1.75 mM EDTA, 0.05% digitonin, 1 mM DTT, and indicated ATP. Reactions were carried out at 37 °C for 10 min and stopped by addition of the reagent from assay kit. The mixture was incubated for 30 min at room temperature before the activity was measured by monitoring the increase of absorbance at 620 nm. Nonlinear regression to the Michaelis-Menten equation and data analysis was performed using OriginPro 8.

## Data availability

Atomic coordinate and EM density map of the hPMCA1-NPTN (PDB: 6A69; EMDB: EMD-6987) have been deposited in the Protein Data Bank (http://www.rcsb.org) and the Electron Microscopy Data Bank (https://www.ebi.ac.uk/pdbe/emdb/). Other data are available from the corresponding authors upon reasonable request.

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

## Acknowledgements

We thank Xiaomei Li and Xiaomin Li for technical support at the cryo-EM facility. We thank the Tsinghua University Branch of China National Center for Protein Sciences (Beijing) for providing the cryo-EM facility support. We thank the computational facility support on the cluster of Bio-Computing Platform (Tsinghua University Branch of China National Center for Protein Sciences Beijing) and the "Explorer 100" cluster system of Tsinghua National Laboratory for Information Science and Technology. This work was supported by funds from National Key Basic Research Program (973 Program) of China (2015CB910101), the National Key R&D Program of China (2016YFA0500402), and the National Natural Science Foundation of China (projects 31621092, 31630017, and 31611130036).

## Author contributions

D.G. conceived the project and designed all experiments. D.G., X.C., K.R. and G.Z. performed experiments. D.G., X.C., G.H., J.L. and Q.Z. conducted the cryo-EM analysis.

All authors contributed to data analysis. D.G. built the atomic model and wrote the manuscript. N.Y. helped with editing the paper before submission.

## Additional information

**Competing interests:** The authors declare no competing interests.

