## [Peer Review File · Nature Communications]

Reviewers' comments:

Reviewer #1 (Remarks to the Author):

In this manuscript by Zhou and coworkers the authors describe the cryo-EM structure of human PMCA1 in complex with neuroplastin (NPTN).

The presented structure is at reasonable resolution (approx. 4Å) and represents the first experimentally determined structure of a plasma-membrane Ca^{2+} -ATPase. The work seems to be technically sound and the comprehensive structural interpretation is well supported by good quality figures.

The detection of the (obligatory) neuroplastin subunit as an additional TM-helix in the structure is interesting. The binding site of this "11th TM-helix" is different from those previously observed in the sodium-potassium pump and SERCA complexes with regulatory TM-subunits. The interpretation that the different binding site of the additional TM-subunit indicates a different effect of NPTN/BASI is, however, speculative or requires further clarification. What are the effects or functional consequences of the additional NPTN-TM-helix at this new position?

My major concern with this work is the following:

A key feature of PMCAs is the autoinhibition that can be regulated by calmodulin or acidic phospholipids. Unfortunately, in the structure presented here, both the autoinhibitory domain (located at the C-terminus) as well as the phospholipid-binding domain are completely absent. Without these characteristic domains, the structure resembles SERCA with the exception that hPMCA1 only contains one Ca^{2+} binding site.

The authors interpret their NPTN-hPMCA1 structure to be in an E1-Mg²⁺-like activated state. This claim should be supported by activity assays. These are relatively easy for P-type ATPases and could provide important insight into the activity status of the investigated samples and the PMCA-NPTN complex in general.

Other points:

- p.12, l.249: Reference #35 (Brini et al, 2017) does not support the claim that hPMCA1-NPTN is in an activated state. Please clarify.
- p.5: it would be good to show the peptide coverage of the mass spec analysis (NPTN vs. BASI)
- p.8, l.159-160: please rephrase sentence
- p.9, l.185: should be E1- Ca^{2+} instead of E1-2 Ca^{2+} , as hPMCA1 only contains one Ca^{2+} binding site
- do the authors see any indication for heterotetrameric complexes in their samples?

Reviewer #2 (Remarks to the Author):

In this manuscript, Deshun Gong et al describe the novel structure of a human Ca^{2+} ATPase by cryo-electron microscopy. The first structure of this class gives a clear picture of the core of the molecule, including the transmembrane region and the binding region of an obligate binding partner, NPTN. The authors show how this structure compares with molecules of a similar function, and propose a mechanism of gated access to the calcium binding site.

Although I have some concerns about the display of the data, I believe structure to be of fine quality and high interest to the community, and the interpretation of the data to be quite solid. With the revisions I suggest below, I recommend the manuscript be published in this journal.

I would like to remind the authors that the disordered regions of today are the interesting regions of tomorrow, so I encourage you to show the reader what you have found, even if it is lower resolution than the core of the molecule.

line 98 - "our results confirm that NPTN and BASI are obligatory" - these results demonstrate that they co-purify, not that they are obligatory.

line 110 - "The crystal structure of rabbit SERCA (PDB code: 3W5B) was docked into the density with manual adjustment (Supplementary Fig. 3)" - Supp Fig 3 is a sequence alignment. I cannot find rabbit SERCA docked somewhere in the figures.

line 119 - "In this paper, the structural elucidation is mainly focused on the transmembrane domain with high resolution." - Perhaps, but if you present coordinates of a domain, the reader should at least be able to understand how you arrived at those coordinates. Rigid body fitting? Flexible fitting? What resolution are the features? Also, see comment for Figure Legend 1

line 126 - "... almost all of which traverse the membrane at a tilt angle of approximately 60° (Fig. 1c)" - do you mean a tilt of 30 degrees? A 60 degree tilt would be a very great tilt indeed. The I am assuming that 0 tilt is perpendicular to the membrane for a transmembrane helix.

line 168 - "NPTN traps..." - If NPTN is obligatory, how can it be responsible for trapping the molecule in the E1-Mg-like state?

line 249 - "...is in an activated state." - what is meant by this? Why does an absent domain signify an activated state?

line 254 - " due to the active effect of subunits..." - which active effects by which subunits?

line 514 - "the eluent was concentrated..." - how? also line 518

line 532 - "the total dose rate..." do you mean total dose?

line 548 - what makes a particle good? did you merge consistent classes? Did the particles have particular statistics?

Figure Legend1 - in my view the local resolution panel does not properly reflect the resolution of the structure. The trans-membrane core portion and the P domain seem fine, but the A and N domains look nothing like what you have modelled, and the Ig2 domain is not even shown. You state in the manuscript, (line 115) "The A domain and the N domain were mapped at a lower resolution," the meaning of which is unclear to me. Does this mean you fitted your model to a map filtered to 6Å? If so, that's fine, just say so. And what made you choose 6Å? In the figure, the resolution colour scheme goes from blue to orange, but the majority of domains N and A are white, a colour I cannot clearly identify in the colour scale. 6Å resolution isn't even on the scale, and you clearly have density resolved more poorly than that. This whole image should really be resolution filtered, not just resolution coloured. Filtering would allow the reader to see ALL the features in the map that you have modelled, and immediately know what level of confidence to place in those regions. A program like blocres, from the BSOFIT package, can help you with this.

Minor points - Grammar and such:

line 68: contains, not containing

line 128 - "... are invisible" - perhaps "... are not resolved" - also line 248

line 205 - "The distance between A370 and T110 in hPMCA1..." - this sentence is difficult to follow. Consider redrafting. The Figure makes clear what you are trying to say, but I had trouble with the text.

line 218: PMCAs are... - should read "the class PIIB PMCAs were identified..."

line 267 - "necessitated" should probably read "necessary"

line 502 - 12 l of cells *were* collected

Figure Legend 1 line 435 - "The 4.1-Å map was generated in Chimera" should maybe read "The image of the 4.1-Å map was generated in Chimera"

line 436 - The density of Ig-2 is not visible at this threshold.

Suppl Fig 2 - You say you merged all the "good" classes? What constitutes good?

Reviewer #1 (Remarks to the Author):

In this manuscript by Zhou and coworkers the authors describe the cryo-EM structure of human PMCA1 in complex with neuroplastin (NPTN). The presented structure is at reasonable resolution (approx. 4Å) and represents the first experimentally determined structure of a plasma-membrane Ca^{2+} -ATPase. The work seems to be technically sound and the comprehensive structural interpretation is well supported by good quality figures. The detection of the (obligatory) neuroplastin subunit as an additional TM-helix in the structure is interesting. The binding site of this “11th TM-helix” is different from those previously observed in the sodium-potassium pump and SERCA complexes with regulatory TM-subunits. The interpretation that the different binding site of the additional TM-subunit indicates a different effect of NPTN/BASI is, however, speculative or requires further clarification. What are the effects or functional consequences of the additional NPTN-TM-helix at this new position?

Response:

Please see the response for the comment #1 below.

My major concern with this work is the following:

A key feature of PMCAs is the autoinhibition that can be regulated by calmodulin or acidic phospholipids. Unfortunately, in the structure presented here, both the autoinhibitory domain (located at the C-terminus) as well as the phospholipid-binding domain are completely absent. Without these characteristic domains, the structure resembles SERCA with the exception that hPMCA1 only contains one Ca^{2+} binding site.

1. The authors interpret their NPTN-hPMCA1 structure to be in an E1- Mg^{2+} -like activated state. This claim should be supported by activity assays. These are relatively easy for P-type ATPases and could provide important insight into the activity status of the investigated samples and the PMCA-NPTN complex in general.

Response: Good advice!

a. To investigate the effects or functional consequences of the NPTN on regulating the activity of hPMCA1, detergent screening for obtaining the hPMCA1 alone proteins was performed during the purification. The subunits were dissociated from the hPMCA1 by washing with dodecyltrimethylammonium chloride (DTAC)-containing buffer (Fig. 2e). The hPMCA1 alone proteins still exhibited good solution behaviour (Supplementary Fig. 6). Accordingly, the ATPase activities of hPMCA1-NPTN and hPMCA1 alone proteins were examined. The K_m and V_{max} for the ATPase activity of hPMCA1-NPTN proteins were measured to be $\sim 519.5 \mu\text{M}$ and $\sim 325.5 \text{ nmol mg}^{-1} \text{ min}^{-1}$, respectively. The hPMCA1 alone proteins were devoid of ATPase activity (Fig. 2f). These results indicate that the hPMCA1-NPTN proteins are functional and the subunits are required for the activity of hPMCA1. We have added the results, methods, and discussion about this part in our revised manuscript.

b. The structure of hPMCA1-NPTN determined in present work is in an intermediate state in the presence of EDTA. The conformation will change if ATP, Mg^{2+} , and Ca^{2+} are added in the sample for measuring the ATPase activity. The release of free Pi is coupled with the transport of Ca^{2+} , which means that the reaction cycle that existence of multiple functional states has to be finished. Accordingly, it is challenging to determine whether an intermediate state is activated or not by activity assay. In addition, the E1- Mg^{2+} state is defined by the crystal structures without evidence of activity assay (Winther AM et al, *Nature*, **495**, 265-269 (2013); Toyoshima C et al, *Nature*, **495**, 260-264 (2013)). Taken together, we can prove that the hPMCA1-NPTN proteins are functional, but I am afraid that we can't determine whether the structure of hPMCA1-NPTN determined in present work is an E1- Mg^{2+} -like activated state or not by activity assay. Fortunately, to date, the crystal structures of SERCA in all major states of the reaction cycle are available, the structural comparison with SERCA can provide important insights into the state of hPMCA1-NPTN. For the accuracy of description, we have revised the subheading "The NPTN traps the calcium pump in an E1- Mg^{2+} like state" to "The hPMCA1 closely resembles to the E1- Mg^{2+} structure" in

the “Results” section.

2. p12, l.249: Reference #35 (Brini et al, 2017) does not support the claim that hPMCA1-NPTN is in an activated state. Please clarify.

Response: Good advice!

We are sorry about this inaccurate statement. We have removed this sentence from our revised manuscript.

3. p.5: it would be good to show the peptide coverage of the mass spec analysis (NPTN vs. BASI)

Response: Good advice!

We have added the results of mass spec as “Supplementary Table 1”, which including the number of unique peptides and peptide coverage of NPTN and BASI.

4. p.8, l.159-160: please rephrase sentence

Response: Good advice!

We have rewritten this sentence as “The TM1 of hPMCA1 is sharply bent as that of E1-Mg²⁺ structure, and the TM2, TM3, TM5, TM6, TM8, and TM9 of hPMCA1 are well aligned with that of E1-Mg²⁺ structure.”

5. p.9, l.185: should be E1-Ca²⁺ instead of E1-2Ca²⁺, as hPMCA1 only contains one Ca²⁺ binding site

Response: Good advice!

We have revised it.

6. do the authors see any indication for heterotetrameric complexes in their samples?

Response:

No indication for the heterotetrameric complexes is found in the samples. In the purification step, the complexes exhibited a single, sharp, and symmetric peak on a size exclusion column, generally indicating that the sample appears to be

monodispersed (Supplementary Fig.1a). Furthermore, no class for heterotetrameric complexes was found after 2D classification during the data processing (Supplementary Fig.1b).

Reviewer #2 (Remarks to the Author):

In this manuscript, Deshun Gong et al describe the novel structure of a human Ca^{2+} ATPase by cryo-electron microscopy. The first structure of this class gives a clear picture of the core of the molecule, including the transmembrane region and the binding region of an obligate binding partner, NPTN. The authors show how this structure compares with molecules of a similar function, and propose a mechanism of gated access to the calcium binding site.

Although I have some concerns about the display of the data, I believe structure to be of fine quality and high interest to the community, and the interpretation of the data to be quite solid. With the revisions I suggest below, I recommend the manuscript be published in this journal.

1. I would like to remind the authors that the disordered regions of today are the interesting regions of tomorrow, so I encourage you to show the reader what you have found, even if it is lower resolution than the core of the molecule.

Response: Good advice!

We appreciate this valuable comment. Usually, the disordered regions play important roles in regulating the function. In our case, although the C-terminal autoinhibition domain as well as the phospholipid-binding domain are completely absent in our structure, both the domains are vital to regulate the pump activity. According to the reviewer's comment, we have added a new figure as "Supplementary Figure 7" to describe the location of the phospholipid-binding domain and discussed the possible effect on the Ca^{2+} access channel in the "Discussion" section.

2. line 98 - "our results confirm that NPTN and BASI are obligatory" - these results

demonstrate that they co-purify, not that they are obligatory.

Response: Good advice!

We have removed this sentence from our revised manuscript.

3. line 110 - "The crystal structure of rabbit SERCA (PDB code: 3W5B) was docked into the density with manual adjustment (Supplementary Fig. 3)" - Supp Fig 3 is a sequence alignment. I cannot find rabbit SERCA docked somewhere in the figures.

Response: Good advice!

We are sorry about this confusion. We have revised it as "The main chain of these regions was built by homology modeling based on the crystal structure of SERCA (PDB: 3W5B) and the side chains were assigned mainly by bulky residues such as Phe, Tyr, Trp, and Arg (Supplementary Fig. 4a)."

4. line 119 - "In this paper, the structural elucidation is mainly focused on the transmembrane domain with high resolution." - Perhaps, but if you present coordinates of a domain, the reader should at least be able to understand how you arrived at those coordinates. Rigid body fitting? Flexible fitting? What resolution are the features? Also, see comment for Figure Legend 1

Response:

In the beginning of this paragraph, we have described the local resolution of these regions and how to build the atomic model as follows: "The EM map is well resolved for most of the P domain and the transmembrane domain, where the local resolution reaches 3.9 Å (Fig. 1a). The main chain of these regions was built by homology modeling based on the crystal structure of SERCA (PDB: 3W5B) and the side chains were assigned mainly by bulky residues such as Phe, Tyr, Trp, and Arg (Supplementary Fig. 4a)."

5. line 126 - "... almost all of which traverse the membrane at a tilt angle of approximately 60° (Fig. 1c)" - do you mean a tilt of 30 degrees? A 60 degree tilt would be a very great tilt indeed. The I am assuming that 0 tilt is perpendicular to the

membrane for a transmembrane helix.

Response: Good advice!

We are sorry about this inaccurate description. It is really true as the reviewer suggested that not all of TMs of hPMCA1 traverse the membrane at a same tilt angle, so we have revised this sentence as “The NPTN-TM traverses the membrane with a tilt angle of approximately 30° (Fig. 1c).” to describe only the single transmembrane helix of NPTN.

6. line 168 - "NPTN traps..." - If NPTN is obligatory, how can it be responsible for trapping the molecule in the E1-Mg-like state?

Response: Excellent concern!

It is really true as the reviewer suggested that the word “trap” is definitely incorrect. We have revised it as “NPTN-bound calcium pump” throughout the manuscript.

7. line 249 - "...is in an activated state." - what is meant by this? Why does an absent domain signify an activated state?

Response: Excellent concern!

We are sorry about this inaccurate statement. The absence of autoinhibitory domain is not enough to claim that the pump is in an activated state. So we have removed this sentence from our revised manuscript. In addition, we have performed the ATPase activities assays to demonstrate that the hPMCA1-NPTN is functional according to another reviewer’s suggestion.

8. line 254 - " due to the active effect of subunits..." - which active effects by which subunits?

Response: Good advice!

We have revised it as “due to the efficiency of PMCA-mediated Ca²⁺ clearance can be improved by the NPTN”.

9. line 514 - "the eluent was concentrated..." - how? also line 518

Response:

We have updated it as “The eluent was concentrated with a 100-kDa cut-off Centricon (Millipore)...”

10. line 532 - "the total dose rate..." do you mean total dose?

Response: Good advice!

Yes, it should be “total dose”, we have corrected it.

11. line 548 - what makes a particle good? did you merge consistent classes? Did the particles have particular statistics?

Response:

After the local angular search 3D classification, classes with the smallest value of the accuracy of rotation and translation were considered as good classes. Yes, we combined the consistent classes from different cycles of local angular search 3D classification and removed the duplicated particles. For clarity, we have revised the sentence “A total of 655,998 good particles were selected from the local angular search 3D classification” to “A total of non-duplicated 655,998 particles were selected from the good classes of local angular search 3D classification, which usually have the smallest value of the accuracy of rotation and translation.”.

12. Figure Legend1 - in my view the local resolution panel does not properly reflect the resolution of the structure. The trans-membrane core portion and the P domain seem fine, but the A and N domains look nothing like what you have modelled, and the Ig2 domain is not even shown. You state in the manuscript, (line 115) "The A domain and the N domain were mapped at a lower resolution," the meaning of which is unclear to me. Does this mean you fitted your model to a map filtered to 6Å? If so, that's fine, just say so. And what made you choose 6Å? In the figure, the resolution colour scheme goes from blue to orange, but the majority of domains N and A are white, a colour I cannot clearly identify in the colour scale. 6Å resolution isn't even on the scale, and you clearly have density resolved more poorly than that. This whole

image should really be resolution filtered, not just resolution coloured. Filtering would allow the reader to see ALL the features in the map that you have modelled, and immediately know what level of confidence to place in those regions. A program like blocres, from the BSOFIT package, can help you with this.

Response: Good advice!

We are sorry about this confusing figure. According to the reviewer's suggestion, we have updated the Figure 1a. Now, the local resolution panel can properly reflect the resolution of the structure. We have changed the threshold of EM map to a proper level for better showing the densities of N domain, A domain, and Ig2 domain. For clarity, we have revised the sentence "The A domain and the N domain were mapped at a lower resolution" to "The densities for the A domain and the N domain were of lower resolutions. Predicted structures for these two domains generated in Phyre2 were docked into the map with manually adjustment (Fig. 1a and Supplementary Fig. 4b).". The EM map low-pass-filtered to 6Å resolution was aimed to dock the crystal structure of Ig2 domain. Because the orientation of Ig2 domain can be determined at 6Å resolution, we choose it.

13. line 68: contains, not containing

Response:

We have corrected it.

14. line 128 - "... are invisible" - perhaps "... are not resolved" - also line 248

Response:

We have revised it.

15. line 205 - "The distance between A370 and T110 in hPMCA1..." - this sentence is difficult to follow. Consider redrafting. The Figure makes clear what you are trying to say, but I had trouble with the text.

Response: Good advice!

We have rewritten this sentence as "In the E2 state of SERCA, which the Ca²⁺ entry

pathway is blocked, the distance between the C α atoms of G257 and L61 is 6 Å. Correspondingly, the distance between the C α atoms of A370 and T110 in hPMCA1-NPTN increases to 16 Å.”.

16. line 218: PMCAs are... - should read "the class PIIB PMCAs were identified..."

Response:

We have revised it.

17. line 267 - "necessitated" should probably read "necessary

Response:

We have revised it.

18. line 502 - 121 of cells “were” collected

Response:

We have revised it.

19. Figure Legend 1 line 435 - "The 4.1-Å map was generated in Chimera" should maybe read "The image of the 4.1-Å map was generated in Chimera"

Response:

We have revised it.

20. line 436 - The density of Ig-2 is not visible at this threshold.

Response:

We have revised it.

21. Suppl Fig 2 - You say you merged all the "good" classes? What constitutes good?

Response:

Same as the question 11. For clarity, we have added a sentence in the “Image processing” section as follows,

“A total of non-duplicated 655,998 particles were selected from the good classes of

local angular search 3D classification, which usually have the smallest value of the accuracy of rotation and translation.”.

REVIEWERS' COMMENTS:

Reviewer #1 (Remarks to the Author):

In the revised version of this manuscript the authors have taken into account most of the suggestions of the referees. The manuscript has improved now.

However, before acceptance, there are still a few points that should be corrected or clarified.

- I strongly disagree with the statement that the hPMCA1 alone proteins obtained from detergent screening (to dissociate the subunits) exhibit "good solution behaviour" (line 161). The new Suppl. Fig. 6 shows a large void peak followed by a peak that probably corresponds to oligomeric hPMCA1 (typical for many PMCA). In my opinion the smaller peak at 13ml is the monomeric hPMCA1 peak, while the two first peaks contain non-functional protein. If this protein was used for activity assays, it is not surprising that hPMCA1 alone does not show any activity.

- I would suggest changing the manuscript as follows: Subunit dissociation leads to oligomerization/aggregation of the protein and subsequent loss of activity. I believe that the hPMCA1-NPTN complex is functional. Without the NPTN subunit, the protein seems to be unhappy. The reason for this is not the focus of this manuscript and the structure of the complex presented here is fine.

- line 70: reference (Brodin 1992) is not included in reference list. The 40 basic amino acids exist in mammalian PMCA, not in all PMCA.

- Suppl. Fig. 2 / Image processing section (line 373): In response to referee #2 this sentence has been added: "A total of non-duplicated 655,998 particles were selected from the good classes of local angular search 3D classification, which usually have the smallest value of the accuracy of rotation and translation." However, this sentence still doesn't explain what constitutes a "good" class. How can a good class have the smallest accuracy?!

- many of the new sentences added to the manuscript need grammar correction: e.g.

- line 218: "in which the Ca²⁺ entry pathway..."

- line 269: "as the efficiency..."

Reviewer #2 (Remarks to the Author):

I am generally satisfied by the corrections to the manuscript. Although the authors opted not to do local resolution-based filtering, they did alter the threshold for Fig 1a, such that the more poorly resolved domains are visible. This satisfies my main criticism for the display of their data.

The correction at line 269 does not make sense to me. Consider revising.

I recommend publication.

Reviewer #1 (Remarks to the Author):

In the revised version of this manuscript the authors have taken into account most of the suggestions of the referees. The manuscript has improved now.

Thanks!

However, before acceptance, there are still a few points that should be corrected or clarified.

- I strongly disagree with the statement that the hPMCA1 alone proteins obtained from detergent screening (to dissociate the subunits) exhibit “good solution behaviour” (line 161). The new Suppl. Fig. 6 shows a large void peak followed by a peak that probably corresponds to oligomeric hPMCA1 (typical for many PMCA). In my opinion the smaller peak at 13ml is the monomeric hPMCA1 peak, while the two first peaks contain non-functional protein. If this protein was used for activity assays, it is not surprising that hPMCA1 alone does not show any activity.

- I would suggest changing the manuscript as follows: Subunit dissociation leads to oligomerization/aggregation of the protein and subsequent loss of activity. I believe that the hPMCA1-NPTN complex is functional. Without the NPTN subunit, the protein seems to be unhappy. The reason for this is not the focus of this manuscript and the structure of the complex presented here is fine.

Response: Excellent concern!

We are sorry about the figure legend of Suppl. Fig. 6 does not provide enough information. Actually, the hPMCA1 alone proteins was purified by Superdex 200 column during last-step purification (Suppl. Fig. 6a) (described in the “Methods” section), while the hPMCA1-NPTN complex was purified by Superose 6 column (Suppl. Fig. 1a). The molecular weight standard for the protein with an elution volume of 13.5 ml (The third peak) on Superdex 200 column is about 75 kDa. Membrane proteins can run at molecular weights higher than would be predicted due to detergent molecules may potentially expand the hydrodynamic radius. The third peak may belong to the chaperone HSP70. The second peak is the monomeric hPMCA1 peak (Suppl. Fig. 6b), the proteins with an elution volume of 11 ml were

used for activity assay. Although the UV absorption at 8.5 ml (void peak) is similar with that at 11 ml (hPMCA1 peak) (Suppl. Fig. 6a), the band at 8.5 ml is much weaker than that at 11 ml (Suppl. Fig. 6b), indicating that the void peak contains large amounts of contaminants that can absorb UV, such as nucleic acids. Although the void peak was not well separated from the hPMCA1 peak, most hPMCA1 proteins that used for activity assay were well folded. For clarity, we have added the column used for purification in the figure legends of Suppl. Fig. 1 and Suppl. Fig. 6. We have added the molecular weight standard in the Suppl. Fig. 6a. We have revised the sentence “The hPMCA1 alone proteins still exhibited good solution behavior” as “Most of the hPMCA1 alone proteins were still well folded.”

- line 70: reference (Brodin 1992) is not included in reference list. The 40 basic amino acids exist in mammalian PMCA, not in all PMCA.

Response: Good advice!

We have updated the reference list and revised the “PMCA” as “mammalian PMCA”.

- Suppl. Fig. 2 / Image processing section (line 373): In response to referee #2 this sentence has been added: “A total of non-duplicated 655,998 particles were selected from the good classes of local angular search 3D classification, which usually have the smallest value of the accuracy of rotation and translation.” However, this sentence still doesn't explain what constitutes a “good” class. How can a good class have the smallest accuracy?!

Response:

The process of 3D classification can separate fully intact particles from incomplete, truncated or fragmented particles and contaminated particles. 3D classification can also separate the particles that contribute negatively to the reconstruction process from those that do contain useful high-resolution information. The class with smallest value of the accuracy of rotation and translation usually has the highest-resolution EM map, which means that the orientation angles of the particle images are best matched

with the reference projections. For clarity, we have revised it as “The class with fully intact particles was considered as a good class, which contains useful high-resolution information and usually has the smallest value of the accuracy of rotation and translation. A total of non-duplicated 655,998 particles were selected from the good classes of local angular search 3D classification.”

- many of the new sentences added to the manuscript need grammar correction: e.g.
- line 218: “in which the Ca²⁺ entry pathway...”
- line 269: “as the efficiency...”

Response:

We have carefully checked the grammar and made corrections.

Reviewer #2 (Remarks to the Author):

I am generally satisfied by the corrections to the manuscript. Although the authors opted not to do local resolution-based filtering, they did alter the threshold for Fig 1a, such that the more poorly resolved domains are visible. This satisfies my main criticism for the display of their data.

Thanks!

The correction at line 269 does not make sense to me. Consider revising.

Response:

We have revised it as “As the NPTN is required for the hPMCA1 functional activity Fig. 2f),”

I recommend publication.

Thanks!